# Castration-Resistant Prostate Cancer Outcome Prediction Using Phased Long Short-Term Memory with Irregularly Sampled Serial Data

**Jihwan Park [1,2], Mi Jung Rho [2], Hyong Woo Moon [3] and Ji Youl Lee [3,*]**

[1]  Department of Biomedicine & Health Sciences, College of Medicine, The Catholic University of Korea,
    Seoul 06591, Korea; bosoagalaxy@gmail.com
[2]  Catholic Cancer Research Institute, College of Medicine, The Catholic University of Korea,
    Seoul 06591, Korea; romy1018@naver.com
[3]  Department of Urology, Seoul St. Mary's Hospital, College of Medicine, the Catholic University of Korea,
    Seoul 06591, Korea; aspasias@catholic.ac.kr
*  Correspondence: uroljy@catholic.ac.kr

**Abstract:** It is particularly desirable to predict castration-resistant prostate cancer (CRPC) in prostate cancer (PCa) patients, and this study aims to predict patients' likely outcomes to support physicians' decision-making. Serial data is collected from 1592 PCa patients, and a phased long short-term memory (phased-LSTM) model with a special module called a "time-gate" is used to process the irregularly sampled data sets. A synthetic minority oversampling technique is used to overcome the data imbalance between two patient groups: those with and without CRPC treatment. The phased-LSTM model is able to predict the CRPC outcome with an accuracy of 88.6% (precision-recall: 91.6%) using 120 days of data or 94.8% (precision-recall: 96.9%) using 360 days of data. The validation loss converged slowly with 120 days of data and quickly with 360 days of data. In both cases, the prediction model takes four epochs to build. The overall CPRC outcome prediction model using irregularly sampled serial medical data is accurate and can be used to support physicians' decision-making, which saves time compared to cumbersome serial data reviews. This study can be extended to realize clinically meaningful prediction models.

**Keywords:** prostate cancer; castration-resistant prostate cancer; deep learning; phased long short-term memory

## 1. Introduction

Prostate cancer (PCa) is one of the most common male cancers. There are more than 3.3 million men suffering from PCa in the USA [1], and its incidence is increasing in Korea [2]. Recently, a national project in Korea has been developing artificial intelligence medical software for several diseases including PCa [3].

Androgen deprivation therapy (ADT) is commonly used to treat PCa patients. However, this fails patients that are classified as having castration-resistant prostate cancer (CRPC). According to the National Cancer Institute (NCI) of the National Institute of Health (NIH), CRPC is a "prostate cancer that keeps growing even when the amount of testosterone in the body is reduced to very low levels. Many early-stage prostate cancers need normal levels of testosterone to grow, but castration-resistant prostate cancers do not." It is important to administer appropriate treatments at the right time. When a patient has appropriate treatment, it will reduce the time and pain from the treatment such as chemotherapy. However, it is a complicated problem to make a treatment decision from cumbersome medical data, which has many irregularly sampled features. To overcome this situation, a CRPC

prediction model can support physicians' decision-making and will help to identify appropriate treatments for PCa patients.

At present, PCa outcomes are usually predicted using nomograms [4–7]. Recently, researchers have begun to explore the use of deep learning methods, which show some promise [8–11], but the majority of these techniques rely on image segmentation and classification [12,13]. Data from longitudinal electronic health records (EHRs) that are irregularly sampled, such as PCa data, does not fit well with deep learning prediction models. Therefore, in this study, we adopt a phased long short-term memory (phased-LTSM) method with the aim of developing a model that can predict the outcome of PCa patients from asynchronously sampled data for various variables using deep learning [14]. In particular, we will develop a model that can identify CPRC patients.

## 2. Materials and Methods

### 2.1. Materials

Data from 1592 patients who had undergone prostatectomy were used to predict the use of CRPC medication. The data consisted of clinical stage information, laboratory data, and treatment events. In total, 8 patients received CRPC medication and 1584 patients did not. We used the receipt of CPRC medication as the binary classification condition. These medications included docetaxel, cabazitaxel, abiraterone, enzalutamide, and radium-223 (see Table 1).

**Table 1.** Demographics of prostate cancer patients.

| Category | Non-Castration-Resistant Prostate Cancer (CRPC) Medication | CRPC Medication | *p*-Value (chi-Squared Test) |
|---|---|---|---|
| Number of cases (*n* = 1592 cases) | 1584 | 8 | |
| Age at diagnosis (%) | | | 0.253 |
| <40 | 1 (0.1) | 0 (0.0) | |
| 40–44 | 3 (0.2) | 0 (0.0) | |
| 45–49 | 16 (1.0) | 0 (0.0) | |
| 50–54 | 54 (3.4) | 0 (0.0) | |
| 55–59 | 165 (10.4) | 2 (25.0) | |
| 60–64 | 303 (19.1) | 1 (12.5) | |
| 65–69 | 461 (29.1) | 0 (0.0) | |
| 70–74 | 429 (27.1) | 2 (25.0) | |
| 75–80 | 146 (9.2) | 3 (37.5) | |
| 80–84 | 6 (0.4) | 0 (0.0) | |
| Mean | 63.64 | 66.5 | |
| T-stage (%) | | | 0.008 |
| T1 | 1 (0.1) | 0 (0.0) | |
| T1a | 7 (0.4) | 0 (0.0) | |
| T1c | 3 (0.2) | 0 (0.0) | |
| T2 | 100 (6.3) | 0 (0.0) | |
| T2a | 141 (8.9) | 0 (0.0) | |
| T2b | 56 (3.5) | 0 (0.0) | |
| T2c | 746 (47.1) | 0 (0.0) | |
| T3 | 4 (0.3) | 0 (0.0) | |
| T3a | 312 (19.7) | 3 (37.5) | |
| T3b | 195 (12.3) | 4 (50.0) | |
| T3c | 1 (0.1) | 0 (0.0) | |
| T4 | 14 (0.9) | 1 (12.5) | |
| Tx | 4 (0.3) | 0 (0.0) | |
| N-stage (%) | | | 0.305 |
| N0 | 379 (23.9) | 3 (37.5) | |
| N1 | 68 (4.3) | 1 (12.5) | |
| Nx | 1137 (71.8) | 4 (50.0) | |
| M-stage (%) | | | 0.422 |
| M0 | 304 (19.2) | 3 (37.5) | |
| M1 | 3 (0.2) | 0 (0.0) | |
| Mx | 1277 (80.6) | 5 (62.5) | |

CRPC patients are treated with specific treatments including chemotherapy, such as docetaxel [15], cabazitaxel [16], abiraterone [17], enzalutamide [18], and radium-223 [19]. These CRPC treatments are delivered in many different combinations alongside ADT medications such as leuprorelin, bicalutamide, goserelin, and degarelix.

We preprocessed the serial data into three-dimensional (3D) vectors to use as the input for the deep learning model. Seven features were used: two laboratory data, prostate specific antigen (PSA) and PSA doubling time (DT), the use of radiation therapy (RT), and four ADT medications—leuprorelin, bicalutamide, goserelin, and degarelix. The timesteps were the days when the features were observed. However, not all of the features are observed at every interval and each observation has irregularly sampled follow-up data (see Figure 1). Because PSA is commonly examined lab data through all of the patient's follow up periods, PSA is the most important factor. PSADT shows the characteristic of PSA kinetics.

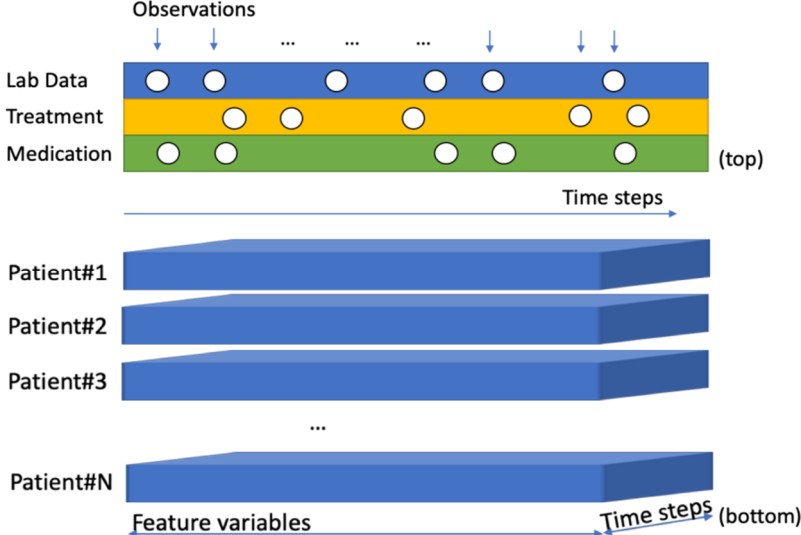

**Figure 1.** Irregularly collected events observations (top) 3D format input for serial data model (bottom).

## 2.2. Ethics

This study procedures were carried out in accordance with the Declaration of Helsinki and were approved by the Institutional Review Board of the Catholic University (IRB number: KC18SNDI0512). Identifying information was removed from the electronic health record (EHR) data, which was then formatted. Unless locally managed mapping information was exposed, specific patients could not be identified.

## 2.3. Methods

We used a 3D fixed time format to input serial data into the model. For the serial data modeling, a recurrent neural network (RNN) managed the data using a circular network model. Simple RNNs have a vanishing gradient problem, which cannot generate proper gradient values from repeated activation functions. The gradient value is an operator that directs the RNN's learning. The activation function calculates the gradient value for each iteration and will produce a value close to 0 after circular network executions. In contrast, long short-term memory (LSTM) uses a gating method to adjust the signal, which eliminates the vanishing gradient problem (see Figure 2). The input vector adjusts the value based on the tanh activation function. The forget vector adjust part of the previous input value to the current input. The output vector adjusts the output values using the previous input and current adjusted input. After all of these gating methods, the final output is used as the next input for the tanh and sigmoid layers.

Typically, the clinical event observations were not collected regularly. This irregular sampling meant that the intervals between consecutive events were not evenly spaced. LSTM cannot learn effectively from such data.

Phased-LSTM selectively updates the status to account for irregular sampling times through an additional time gate [14]. At each time step, the time gate switches on and off to update the state and

the hidden output state. These updates only happen for the open phase. Otherwise, the previous values will be used. The time gate addresses the LSTM problems caused by irregularly sampled data (see Figure 3).

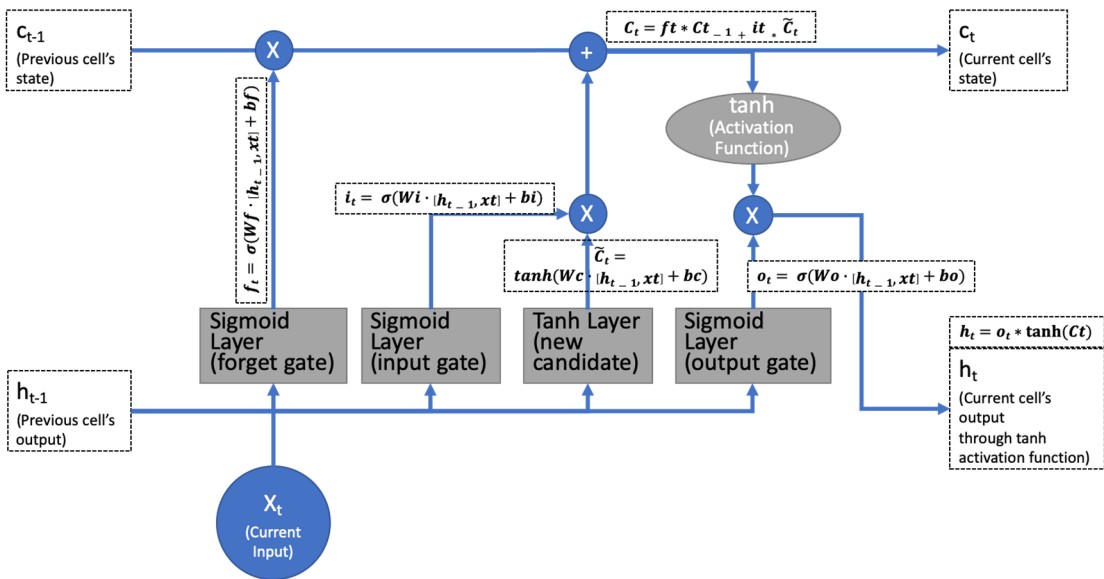

**Figure 2.** Standard long short-term memory (LSTM) using three gates (forget, input, and output) to eliminate the vanishing gradient problem.

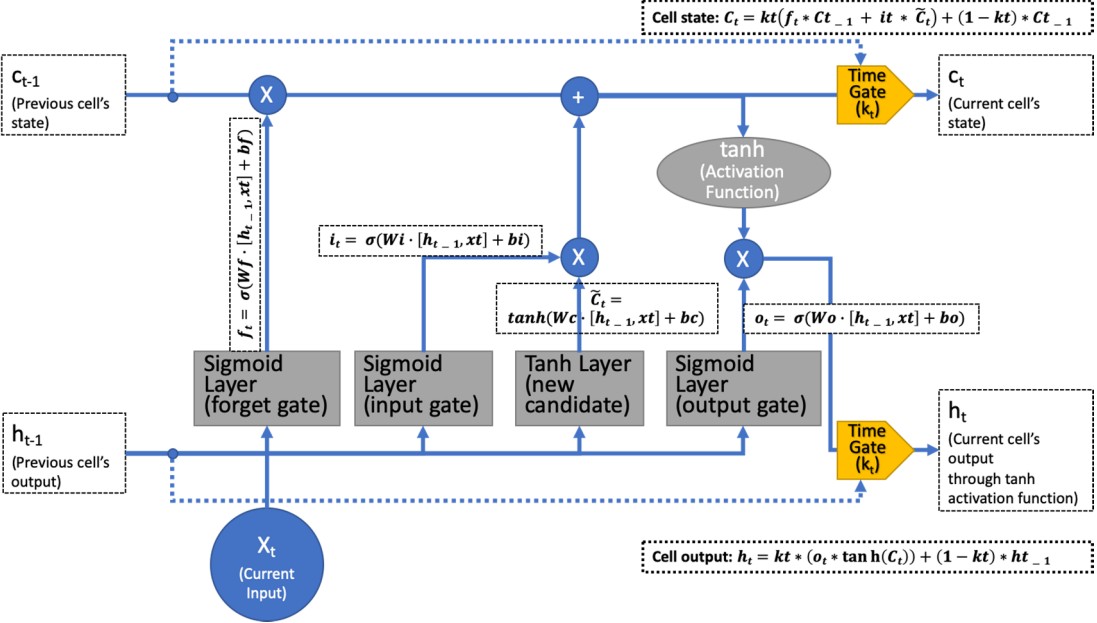

**Figure 3.** Phased LSTM using rectified linear units as the time-gate kt that propagates the gradient well.

There are two patterns of serial medical data: in the first, more than two features are sampled simultaneously and, in the second, some parts of the features are missing and others are not. To manage both patterns, missing imputation must be applied. The phased-LSTM's time gate addresses these patterns based on three phases (see Figure 4).

The time gate of phased-LSTM $k_t$ has three phases: phase 1, "openness-rise" 0 to 1, phase 2, "openness-drop" 1 to 0, and phase 3, "closed" a leak at a rate $\alpha$ [14].

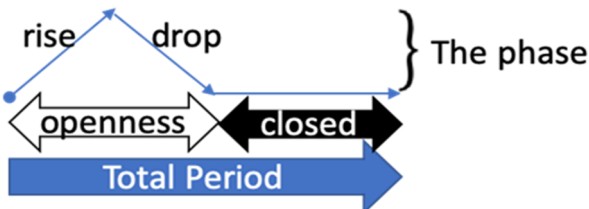

**Figure 4.** Phased LSTM time gate: phase visualization [14].

The time gate $k_t$ is described by

$$\phi_t = \frac{(t-s) \, mod \, \tau}{\tau} \tag{1}$$

and

$$k_t \begin{cases} \frac{2\phi_t}{r_{on}}, & if \; \phi_t < \frac{1}{2} r_{on} \\ 2 - \frac{2\phi_t}{r_{on}}, & if \; \frac{1}{2} r_{on} < \phi_t < r_{on} \\ \alpha \phi_t, & otherwise \end{cases} \tag{2}$$

In this case, $\phi_t$ is an auxiliary variable that represents the phase of the cycle (Equation (1)), $t$ is the time point, $s$ is the phase shift, $\tau$ is the period that controls the real-time period, and $r_{on}$ is the ratio of the open period to the total period $\tau$ [14].

Phased-LSTM's time gate $k_t$ is a kind of rectified linear unit that can propagate gradient information effectively [14] (Equation (2)). Thus, $k_t$ has three parameters: $\tau$, $s$, and $r_{on}$. These parameters were used during the phased-LSTM model learning. During the open-phase, the rise or drop period in Figure 4 known as propagates new cell states or $\alpha$-weighted gradient information. Ultimately, $k_t$ is able to propagate irregularly sampled data selectively. In other words, it deals with complicated time series medical data effectively. Neil et al. presented this algorithm, which outperforms previous LSTM algorithms [14].

We used a chi-squared test to find the characteristic of patients' data. The chi-squared result showed the information of group differences, which represent how each of the groups performed in the study [20].

Due to the imbalance in the ratio of CRPC and non-CRPC patients, we used the synthetic minority oversampling technique (SMOTE) algorithm [21]. The SMOTE generates sample data that is similar to a prior set. This method is frequently used for imbalanced data analysis [22–25]. Based on the SMOTE oversampling algorithm, we generated a 50:50 ratio of CRPC data using the prior data.

We used k-fold cross validation to verify that the result of model training was not overfitted. The k-fold cross validation splits the data set into a k number of sets and validates the model with shuffled different sets [26]. We used 10-folds to verify the model.

The phased-LSTM prediction model was built using TensorFlow [27]. The model consists of a a phased-LSTM layer and used a SoftMax [28] dense layer to generate a binary result: CRPC or non-CRPC treated. It also used the Adam [29] optimizer and sparse_categorical_crossentropy loss function [12].

We used linear interpolation to build serial input data for the phased-LSTM model with an appropriate shape. Linear interpolation is one of the most common algorithms used to build connected serial data from data that has been irregularly recorded.

## 3. Results

The majority of age distribution resided between 55 and 80 years. The mean of age is 63.64 and 66.5 years for each Non-CRPC medication and CRPC medication group. The majority of T-stage data distribution resided between T2 and T3. Most of N-stage and M-stage data resided at a 0 or X stage, which means the absence of metastasis. Though it is highly imbalanced, the data distribution showed a similar shape from two groups.

Figure 5 shows the prior data shape, oversampling result, and scaling result for 120 and 360 days of serial data. We used SMOTE to generate CRPC-treated data similar to the prior CRPC cases. Lastly, we obtained balanced data (1575:1576, CRPC: non-CRPC treated) compared to the prior data that was highly imbalanced (8:1584, CRPC: non-CRPC treated).

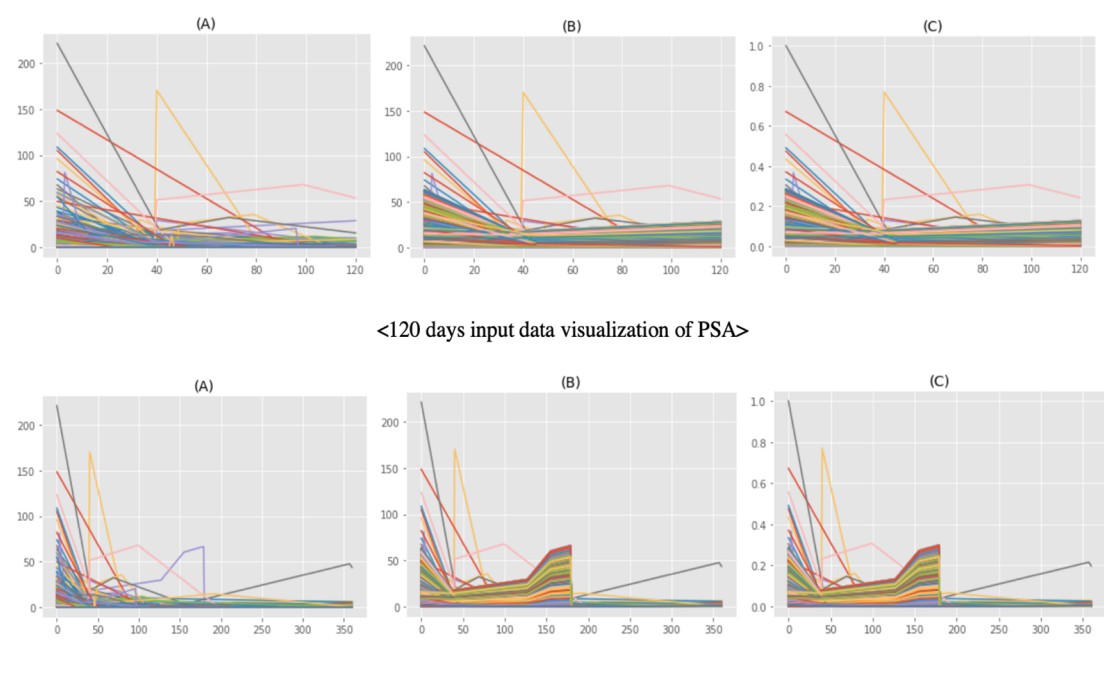

**Figure 5.** Shape of the input data for prostate specific antigens: (**A**) prior data, (**B**) oversampling result from prior data, and (**C**) scaling result from oversampling data.

We used two sets of serial data: 120 days and 360 days. Each set had a plus-one timestep. For instance, in the 120-day case, there were 121 (120 + 1) time steps including the start time point. The 120-day set contained 3152 cases, 121 timesteps, and seven features. The 360-day set contained 3152 cases, 361 timesteps, and seven features. The model parameter number for the model was 70,016, which was trained by the deep learning model.

Both sets used four epochs (blue lines), which is the number of training iterations. To prevent building the overfitted model, we manually determined four epochs by testing 10 epochs (see Table 2). Although the training time was short, the 120-day set has a recall precision of 91.6% and the 360-day set had a recall precision of 96.9%. The validation loss (red lines) of the 120-day set converged slowly, while that of the 360-day set converged quickly (see Figure 6). Ultimately, the Receiver Operating Characteristic/Area Under the Curve (ROCAUC) accuracy result of the 120-day model is 88.6% and the 360-day model is 94.8% (see Figure 7).

Because the loss of the model continued decreasing and the accuracy of the model continued increasing after four epochs, we did 10-fold cross-validation to verify 10 epochs' training results did not overfit (see Table 3). We have 97.81% and 97.68% average accuracy of the model's evaluations every 120 days and 360 days.

**Table 2.** Determining epochs to build model.

| | Running Result: 10 Epochs to Determine Early Stop | | | Running Result: Determine 4 Epochs for Preventing Overfitting | | | |
|---|---|---|---|---|---|---|---|
| Category | Epoch (10) | Loss | Accuracy | Category | Epoch (4) | Loss | Accuracy |
| 120-day model | 1 | 0.6892 | 0.5743 | 120-day model | 1 | 0.6889 | 0.5168 |
| | 2 | 0.6722 | 0.5549 | | 2 | 0.6705 | 0.6954 |
| | 3 | 0.5841 | 0.7257 | | 3 | 0.5720 | 0.7756 |
| | 4 | 0.4445 | 0.8073 | | 4 | 0.3924 | 0.8391 |
| | 5 | 0.2879 | 0.8957 | | | | |
| | 6 | 0.1943 | 0.9379 | | | | |
| | 7 | 0.1496 | 0.9560 | | | | |
| | 8 | 0.1428 | 0.9615 | | | | |
| | 9 | 0.1341 | 0.9637 | | | | |
| | 10 | 0.1140 | 0.9701 | | | | |
| 360-day model | 1 | 0.6848 | 0.5698 | 360-day model | 1 | 0.6882 | 0.4932 |
| | 2 | 0.5190 | 0.7969 | | 2 | 0.5423 | 0.8196 |
| | 3 | 0.2708 | 0.9021 | | 3 | 0.3228 | 0.8930 |
| | 4 | 0.2302 | 0.9180 | | 4 | 0.2388 | 0.9288 |
| | 5 | 0.1519 | 0.9569 | | | | |
| | 6 | 0.1065 | 0.9719 | | | | |
| | 7 | 0.1060 | 0.9742 | | | | |
| | 8 | 0.0861 | 0.9764 | | | | |
| | 9 | 0.0749 | 0.9837 | | | | |
| | 10 | 0.0942 | 0.9782 | | | | |

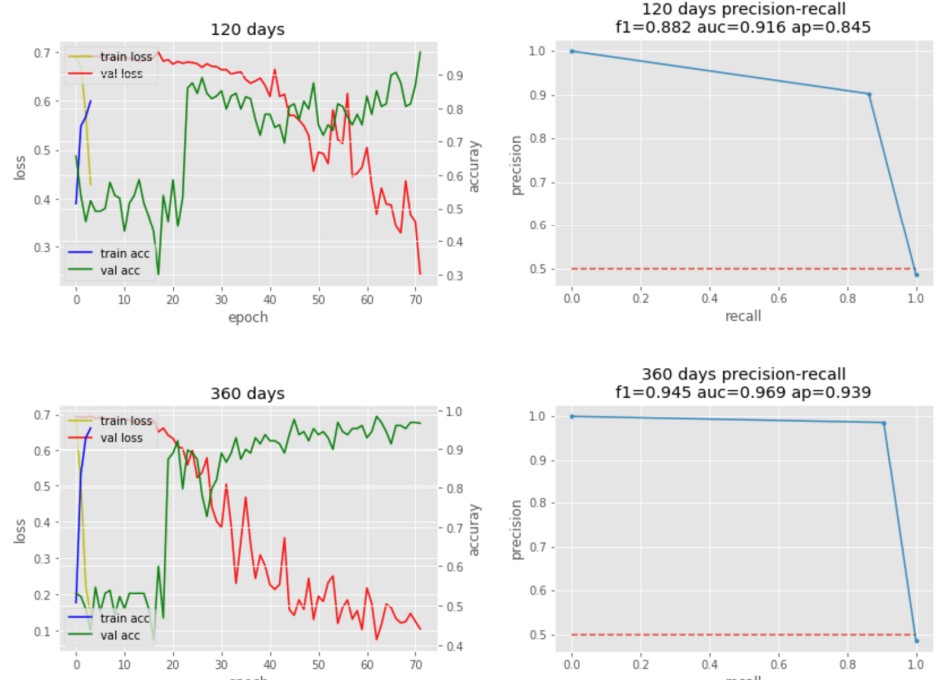

**Figure 6.** Phased LSTM training/validation (left) and precision-recall plots (right).

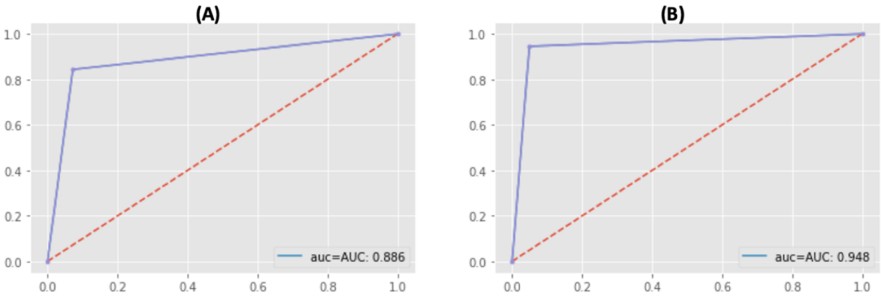

**Figure 7.** Accuracy Result (ROCAUC): (**A**) 120-day model's accuracy is 88.6%, (**B**) 360-day model's accuracy is 94.8%.

**Table 3.** 10-fold cross validation of the model.

| 10-Fold Cross Validation | 120-Day Model | | 360-Day Model | |
|---|---|---|---|---|
| | **Loss** | **Accuracy** | **Loss** | **Accuracy** |
| 1 | 0.1067 | 0.9778 | 0.2015 | 0.9937 |
| 2 | 0.0596 | 0.9810 | 0.0548 | 0.9873 |
| 3 | 0.0750 | 0.9778 | 0.1027 | 0.9714 |
| 4 | 0.1263 | 0.9587 | 0.1008 | 0.9778 |
| 5 | 0.2235 | 0.9841 | 0.0897 | 0.9746 |
| 6 | 0.1063 | 0.9714 | 0.1090 | 0.9746 |
| 7 | 0.1342 | 0.9905 | 0.0859 | 0.9683 |
| 8 | 0.2189 | 0.9937 | 0.0930 | 0.9746 |
| 9 | 0.0863 | 0.9810 | 0.1290 | 0.9651 |
| 10 | 0.1190 | 0.9651 | 0.0744 | 0.9810 |
| Average | 0.1256 | 0.9781 | 0.1041 | 0.9768 |

## 4. Discussion

In this study, we built a model to predict CPRC-treated patients. From this, we were able to draw the following conclusions.

Serial medical data is an important factor in predicting the likely outcome of patients with PCa. For example, predicting the likelihood of CRPC helps medical professionals to make appropriate decisions at the right time. Some previous studies have predicted time or variables related to CRPC. For example, Elishmereni et al. used a mathematical algorithm to predict the time until biochemical failure of ADT using data from 83 hormone sensitive prostate cancer patients [30]. Although this study only used data from 83 patients, it had a prediction accuracy of 90%. In this study, we used data from 1592 patients to predict CRPC, which is a large number of patients for building a prediction model. Another study by Humphreys et al. tried to find a relationship between age at diagnosis and CRPC. They found that those aged between 55 and 75 years have the shortest survival time [31]. We did not use age as an input variable since there was no significant difference (chi-squared test p-value of 0.253) between the groups with and without CRPC treatment (see Table 1). In another study, Angulo et al. used DNA hypermethylation profiles to predict CRPC. They found 61 genes related to CRPC [32]. However, DNA sequencing is not commonly used for PCa patients and it incurs extra expenses for the patient. We used typical follow-up data from PCa patients. A model based on commonly collected data make it more applicable in real out-patient scenarios.

We built a CRPC prediction model, which can support physicians' decision making and allows them to review more patients in a short period of time. When medical staff do not have enough time to track the status of each patient in need of significant interventions, human errors are more likely to occur and opportunities for treatment may be missed until the condition becomes serious. If prediction support is available, such as the likelihood of a PCa patient developing CPRC, patients in need of treatment can be identified quickly. Therefore, patients will receive the right treatment at the right time.

We used the state-of-the-art deep learning method to handle serial medical data. Because it is difficult to predict a patient's outcome with serial data that is collected irregularly, including medications, treatments, and laboratory test results such as blood tests. Typical deep learning methods such as RNN and LSTM can be used to analyze serial data. However, they must be improved in order to handle irregularly-sampled serial datasets. To overcome this, we adopted recently developed deep learning technology. The phased-LSTM [14] deep learning method builds a fast and accurate prediction model using serial medical data. The predictions show high precision recall scores, which means that they are highly accurate.

The results of this study can be extended to other areas such as emergency room or intensive care units, which receive a significant amount of serial data from patients. In addition, follow-up data from many patient care organizations can be used with this method to make accurate predictions that support clinicians' decision-making.

Although we produced meaningful results, this study still has some limitations. We used data from 1592 patients taken from one hospital. Because we were not permitted to use data from other hospitals for this study, we only used one hospital's data for this study. Though the data was from only one hospital, 10 years of follow-up data of 1592 patients is a big enough number to build a prediction model. Because there were many missing values for other features such as family history, genotype, and lifestyle, we built a model without those features.

In addition, we used imbalanced PCa treatment data with only a small number of CRPC-treated patients. Extremely imbalanced data is not enough to verify the model. For instance, when validation data have only a few CRPC patients out of hundreds of PCa patients, it does not make sense to verify the model's accuracy. Chawla et al. proposed SMOTE and presented results with imbalanced data sets, which has 10923 examples in the majority class and 260 examples in the minority class originally [21]. We used the SMOTE method to build oversampling data. Lastly, we got well balanced data, which has a similar shape to the prior data based on SMOTE.

In future studies, we will obtain more CRPC-treated cases in order to improve the model. However, our study is an example of how irregularly sampled serial medical data can be used to build a decision support prediction model.

Ultimately, this study shows how the state-of-the-art phased-LSTM deep learning method can be used to make meaningful prediction models based on serial medical data. This can help clinicians who want to know their patients' likely outcomes from irregularly sampled follow up data.

**Author Contributions:** J.P. wrote the article and helped to develop the CRPC prediction model. M.J.R. supported in writing the article and in developing the CRPC prediction model. H.W.M. gave medical advice for the development of the CRPC prediction model. J.Y.L. supervised the research and tested the CRPC prediction model. All four authors substantially contributed their expertise. All authors have read and agreed to the published version of the manuscript.

**Funding:** The Institute for Information & Communications Technology Promotion (IITP) and grant funded by the Korea government (MSIT) (2018-2-00861, Intelligent SW Technology Development for Medical Data Analysis) funded this research. The Smart Healthcare Research Grant through the Daewoong Foundation (DF-201907-0000001) also supported this research.

**Conflicts of Interest:** The authors declare no conflict of interest.

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
