# Peer review of "Castration-Resistant Prostate Cancer Outcome Prediction Using Phased Long Short-Term Memory with Irregularly Sampled Serial Data"

_applsci, doi:10.3390/app10062000_

Round 1

Reviewer 1 Report

Major comment:

The authors stated that 4 epochs were used without explaining how this number was gotten. Did the authors use early stopping during the model building? If early stopping was used, please submit an additional document of output. Also, accuracy is preferred than precision and recall, could the authors provide additional information on this?

Other comments:

The language may need a bit more work. In introduction, “PCa is one of the biggest part of artificial intelligence medical software development project in Korea lately…” sound awkward, needs revision; also, LSTM is short for long short-term memory, not long-term short-term memory; “castrate-resistant prostate cancers” should be “castration-resistant prostate cancers” in the first paragraph of the introduction; “A CRPC prediction model can support …” this sentence does not flow well from the previous sentences, needs more details before getting to this point.

Author Response

Major comment: (figures are in the attached word file)

The authors stated that 4 epochs were used without explaining how this number was gotten. Did the authors use early stopping during the model building? If early stopping was used, please submit an additional document of output. Also, accuracy is preferred than precision and recall, could the authors provide additional information on this?

Response: Thank you for your kind comment. We included accuracy figure 7 and revised accuracy on this manuscript with precision and recall. We included additional table to determining number of epochs in line from 159-164. Following is the additional information.

Revised contents:

Both sets used 4 epochs (blue lines), which is the number of training iteration. To prevent to build overfitted model, we manually determined 4 epochs from testing 10 epochs (see Table 2). Although the training time was short, the 120 day set has a recall precision of 91.6% and the 360 day set had a recall precision of 96.9%. The validation loss (red lines) of the 120 days set converged slowly, while that of the 360 day set converged quickly (see Fig.6.) Ultimately, the ROCAUC accuracy result of 120 days model is 88.6% and 360 days model is 94.8%. (see Fig.7.)

Table 2. Determining epochs to build model

running result:
10 epochs to determine early stop.

running result:
determine 4 epochs for preventing overfitting

category

epoch(10)

loss

accuracy

category

epoch(4)

loss

accuracy

120 days
model

1

0.6892

0.5743

120 days
model

1

0.6889

0.5168

2

0.6722

0.5549

2

0.6705

0.6954

3

0.5841

0.7257

3

0.5720

0.7756

4

0.4445

0.8073

4

0.3924

0.8391

5

0.2879

0.8957

6

0.1943

0.9379

7

0.1496

0.9560

8

0.1428

0.9615

9

0.1341

0.9637

10

0.1140

0.9701

360 days
model

1

0.6848

0.5698

360 days
model

1

0.6882

0.4932

2

0.5190

0.7969

2

0.5423

0.8196

3

0.2708

0.9021

3

0.3228

0.8930

4

0.2302

0.9180

4

0.2388

0.9288

5

0.1519

0.9569

6

0.1065

0.9719

7

0.1060

0.9742

8

0.0861

0.9764

9

0.0749

0.9837

10

0.0942

0.9782

Other comments:

The language may need a bit more work. In introduction, “PCa is one of the biggest part of artificial intelligence medical software development project in Korea lately…” sound awkward, needs revision; also, LSTM is short for long short-term memory, not long-term short-term memory; “castrate-resistant prostate cancers” should be “castration-resistant prostate cancers” in the first paragraph of the introduction; “A CRPC prediction model can support …” this sentence does not flow well from the previous sentences, needs more details before getting to this point.

Response: We revised introduction part as you mentioned in line from 31-34 and 40-45. We also revised LSTM’s full description as you mentioned. Following is the corrected introduction. To make sure we had professional English editing.

Revised contents:

(line 31-45) Prostate cancer (PCa) is one of the most common male cancers. There are more than 3.3 million men suffering from PCa in the USA [1], and its incidence is increasing in Korea [2]. Recently, national project in Korea has been developing artificial intelligence medical software for several disease including PCa [3].

(line 40-45) Because, when a patient has appropriate treatment, it will reduce the time and pain from the treatment, for instance chemo-therapy. However, it is complicated problem to make treatment decision from cumbersome medical data which has many irregularly sampled features. To overcome this situation, a CRPC prediction model can support physicians’ decision making and will help to identify appropriate treatments for PCa patients.

Certification of English editing is in the attached word fild.

Reviewer 2 Report

In this study, Park and colleagues investigated the accuracy of phased long-term short-term memory (phased-LSTM) model in prediction of patients with prostate cancer who might have castration-resistant prostate cancer (CRPC). They used a time-gate module to minimize the effect of irregularities in the collected data from 1592 patients with prostate cancer (8 with CRPC). To overcome the data imbalance between those with and without CRPC treatment they used the synthetic minority oversampling technique (SMOTe) technique. The authors found that the phased-LSTM model was able to predict the CRPC outcome with 91.6% and 96.9% using 120 and 360 days of data, respectively.

The manuscript and method are well-written; however, there are some considerations as follows:

1. In the application of SMOTE, what ratio of majority class/minority class is acceptable and does only 8 CRPC subjects satisfy vs 1584 PCa patients satisfy the acceptable ratio?

2. The authors only used age and TNM stage for their prediction model and obtained high accuracy in their prediction. Does this mean that other risk factors of prostate cancer such as family history, genotype and lifestyle/diet have less influence on the prediction of CRPC outcome?

3. In the application of SMOTe, it seems that the authors applied the algorithm on the dataset as a whole and then shuffled and splatted the dataset into training and validation sets. Some have criticized this approach for resulting in over-estimation of accuracy and other performance measures of the classifier or the machine learning model. Could the authors compare their results with shuffling and splitting the Dataset into training and validation sets and applying SMOTe on the training dataset and whether they get the same performance and accuracy?

4. Though the authors stated as a limitation, external validation is the key missing part.

5. Could the authors include what features more influenced the performance?

Author Response

In this study, Park and colleagues investigated the accuracy of phased long-term short-term memory (phased-LSTM) model in prediction of patients with prostate cancer who might have castration-resistant prostate cancer (CRPC). They used a time-gate module to minimize the effect of irregularities in the collected data from 1592 patients with prostate cancer (8 with CRPC). To overcome the data imbalance between those with and without CRPC treatment they used the synthetic minority oversampling technique (SMOTe) technique. The authors found that the phased-LSTM model was able to predict the CRPC outcome with 91.6% and 96.9% using 120 and 360 days of data, respectively.

The manuscript and method are well-written; however, there are some considerations as follows:

  1. In the application of SMOTE, what ratio of majority class/minority class is acceptable and does only 8 CRPC subjects satisfy vs 1584 PCa patients satisfy the acceptable ratio?

Response: Thank you for your kind comment. We revised discussion part based on your comment. We had 10 year follow up serial data of total 1592 PCa patients. The case of CRPC subjects is small, however, including data is huge volume of serial data.

Revised part: discussion line from 213-216

Chawla et al. proposed SMOTE and presented results with imbalanced data sets which has 10923 examples in the majority class and 260 examples in the minority class originally [20]. We used the SMOTE method to build oversampling data. Finally, we got well balanced data which has similar shape to the prior data based on SMOTE.

  1. The authors only used age and TNM stage for their prediction model and obtained high accuracy in their prediction. Does this mean that other risk factors of prostate cancer such as family history, genotype and lifestyle/diet have less influence on the prediction of CRPC outcome?

Response: We revised to answer your comment.

Revised part: discussion line from 207-208

Because there were many missing values for other features such as family history, genotype, and lifestyle; we built model without those features.

  1. In the application of SMOTe, it seems that the authors applied the algorithm on the dataset as a whole and then shuffled and splatted the dataset into training and validation sets. Some have criticized this approach for resulting in over-estimation of accuracy and other performance measures of the classifier or the machine learning model. Could the authors compare their results with shuffling and splitting the Dataset into training and validation sets and applying SMOTe on the training dataset and whether they get the same performance and accuracy?

Response: As the same reason your mentioned, Chaswla et al. proposed SMOTE to overcome minor class issue. We harnessed SMOTE to overcome minor class based on Chaswla et al.’s study. They also presented results with imbalanced data sets which has 10923 examples in the majority class and 260 examples

Revised part: discussion line from 210-216

In addition, we used imbalanced PCa treatment data with only a small number of CRPC treated patients. Because extremely imbalanced data is not enough to make models accuracy result, for instance, when validation data have only few CRPC patient out of hundreds PCa patients does not make sense to compare the model’s accuracy. Chawla et al. proposed SMOTE and presented results with imbalanced data sets which has 10923 examples in the majority class and 260 examples in the minority class originally [20]. We used the SMOTE method to build oversampling data. Finally, we got well balanced data which has similar shape to the prior data based on SMOTE.

  1. Though the authors stated as a limitation, external validation is the key missing part.

Response: We revised to answer your comment. We have limited permission to access external medical data in this study.

Revised part: discussion line from 205-207

Because, we were not permitted to use other hospitals data for this study, we only used one hospital’s data for this study. Though the data was from only one hospital, 10 years follow up data of 1592 patients is big enough number to build prediction model.

  1. Could the authors include what features more influenced the performance?

Response: We revised to answer your comment. We can observe Prostate Specific Antigen(PSA) is the only feature which collected through whole follow up period of patients. We used the other features to support the prediction model. That is why the PSA is the most important factor influenced the performance. We also used PSA doubling time(PSADT) which explains characteristics of PSA kinetics.

Revised part: discussion line from 72-74

Because PSA is commonly examined lab data through all of the patient’s follow up periods, PSA is the most important factor. PSADT shows the characteristic of PSA kinetics.

We also had professional English editing:

Certification of English editing is in the attached word file.

Round 2

Reviewer 1 Report

It seems that the author arbitrarily stopped the model training at 4 epochs, because in the submitted file, the loss of the model continued decreasing and the accuracy continued increasing after 4 epochs. Although it is okay for a ~80% accuracy for 120-day model, and ~91% accuracy for 360-day model, if the models are still improving through a bit more epochs, I would not suggest stop training prematurely. In this case, if the cross-validation is applied, overfitting is not overly concerning. 

Author Response

It seems that the author arbitrarily stopped the model training at 4 epochs, because in the submitted file, the loss of the model continued decreasing and the accuracy continued increasing after 4 epochs. Although it is okay for a ~80% accuracy for 120-day model, and ~91% accuracy for 360-day model, if the models are still improving through a bit more epochs, I would not suggest stop training prematurely. In this case, if the cross-validation is applied, overfitting is not overly concerning. 

Answer:

Thank you for your kind comment.  We did 10-folds cross validation.

We added followings to answer your comment.

In the methods ( line 136~138 ) :

We used k-fold cross validation to verify the result of model training was not overfitted. The k-fold cross validation splits data set into k number of sets and validates the model with shuffled different sets[25]. We used 10-folds to verify the model.

In the results ( line 171~176 ) :

Because the loss of model continued decreasing and the accuracy of model continued increasing after 4 epochs, we did 10 folds cross-validation to verify 10 epochs training results were not overfitted (see Table 3). We have 97.81% and 97.68% average accuracy of model’s evaluations of each 120-days and 360-days.

Table 3. 10-folds cross validation of the model

10-folds

cross validation

120-days model

360-days model

loss

accuracy

loss

accuracy

1

0.1067

0.9778

0.2015

0.9937

2

0.0596

0.9810

0.0548

0.9873

3

0.0750

0.9778

0.1027

0.9714

4

0.1263

0.9587

0.1008

0.9778

5

0.2235

0.9841

0.0897

0.9746

6

0.1063

0.9714

0.1090

0.9746

7

0.1342

0.9905

0.0859

0.9683

8

0.2189

0.9937

0.0930

0.9746

9

0.0863

0.9810

0.1290

0.9651

10

0.1190

0.9651

0.0744

0.9810

Average

0.1256

0.9781

0.1041

0.9768

Reviewer 2 Report

The manuscript has been improved in response to the comments. Some of the issues are the limitations of the study which the authors discussed. However, the manuscript still requires text editing, despite that the authors present a certificate for professional English editing. To name a few,  "Because extremely imbalanced data is not enough to make models accuracy result, for instance, when validation data have only few CRPC patient out of hundreds PCa patients does not make sense to compare the model’s accuracy." is hard to follow and understand; Contributions, not Contributionse; data with plural verb not singular verb throughout the manuscript; "It is a complicated problem..." not "it is complicated problem..."; "we built a model without..." not "we built model without..."; "to build a prediction model" not "to build prediction model"; "to use data from other hospitals" not "to use other hospitals data" and etc. 

Author Response

The manuscript has been improved in response to the comments. Some of the issues are the limitations of the study which the authors discussed. However, the manuscript still requires text editing, despite that the authors present a certificate for professional English editing. To name a few,  "Because extremely imbalanced data is not enough to make models accuracy result, for instance, when validation data have only few CRPC patient out of hundreds PCa patients does not make sense to compare the model’s accuracy." is hard to follow and understand; Contributions, not Contributionse; data with plural verb not singular verb throughout the manuscript; "It is a complicated problem..." not "it is complicated problem..."; "we built a model without..." not "we built model without..."; "to build a prediction model" not "to build prediction model"; "to use data from other hospitals" not "to use other hospitals data" and etc. 

Answer:

Thank you for your kind comment. We corrected manuscript as you commented.

Line 221~223:

Extremely imbalanced data is not enough to verify model. For instance, when validation data have only few CRPC patient out of hundreds PCa patients, it does not make sense to verify the model’s accuracy.

Line 233:

Author Contributions:

Line 42:

However, it is a complicated problem to make treatment decision from cumbersome medical

Line 219:

we built a model without those features.

Line 217:

to build a prediction model.

Line 215:

to use data from other hospitals
